

# An integrated tsunami inundation and risk analysis at the Makran Coast, Pakistan

Rashid Haider[1,3], Sajid Ali[1], Gösta Hoffmann [2], Klaus Reicherter[1]

[1] Institute of Neotectonics and Natural Hazards, RWTH Aachen University, Lochnerstr. 4-20, 52056 Aachen, Germany
[2] UNESCO Global Geoparks Unit, Department Heritage, Nature, Society, German Commission for UNESCO, Martin-Luther-Allee 42 53175 Bonn, Germany
[3] Geoscience Advanced Research Laboratories, Geological Survey of Pakistan, Islamabad 45500, Pakistan

*Correspondence to*: Rashid Haider (r.haider@nug.rwth-aachen.de)

**Abstract.** The coastal cities and areas of Gwadar and Pasni in Pakistan, being highly vulnerable to tsunamis, are investigated for inundation and risk analysis. For modelling, both dynamic and static approaches are used for better understanding and for their comparative analysis. The tsunami wave potential in the Arabian Sea is estimated by compiling and assessing the palaeo-morphodynamics of tsunamite record discovered so far along its coastlines. The dynamic model is calibrated and

validated with the inundation limits of the 1945 tsunami in the Pasni area. Further, for risk estimation, three different wave scenarios (7 m, 10 m, and 15 m) are projected on Pasni and Gwadar. Results show that both cities are highly vulnerable to wave heights ≥7 m and wave lengths ≥ 15 km. The 15 m scenario would be almost a complete disaster for both cities and adjoining towns. Results of simulation and comparative analysis also show that due to coastal orientation and morphology, the reflection, integration, and amplification phenomena have a devastating effect on the region, and their intensity increases

with the increasing size of the approaching waveform. It variably but significantly affects the inundation extents and depths. At the end, based on the simulated results, the utility of the installed early warning system is assessed.

## 1 Introduction

Tsunamis are rare events but have high damage potential for the coastal communities. In the past 27 years, 64 tsunamis with a run-up higher than 3 m have been documented globally (Tarbotton et al., 2015). In the recent two decades, there has been a

substantial advancement in our understanding of tsunami formation, propagation, and inundation, enabling planners, administrators, and engineers to calibrate the building's design codes in order to mitigate the disaster risk (Park et al., 2013; Macintosh, 2013). In this regard, inundation analysis is an important tier in vulnerability and risk assessment. It is done by integrating the fragility curves (damage probability curves) with different tsunami-flow magnitudes (Rehman and Cho, 2016). The fragility curves are generated through various techniques based on engineering and statistical approaches, while

the tsunami-flow magnitude is a product of flow depth and flow velocity calculated through dynamic inundation modelling.

In this paper, two aspects have been investigated; primarily the inundation analysis and secondarily a preliminary risk estimation for the coastal cities of Pasni and Gwadar (see Fig. 1). These cities were worst affected by the 1945 earthquake



(Mw 8.1) entailing tsunami which resulted in hundreds of casualties (Pendse, 1946; Hoffmann et al., 2013b). In 2013, a small tsunami (Mw 7.7) struck the same area once more but caused no damage (Heidarzadeh and Satake, 2014b). Both cities had grown up many times as compared to those in 1945, and in the case of any such disaster, a huge life and property losses are expected. Kakar et al. (2014) under the patronage of the UNESCO-Intergovernmental Oceanographic Commission (IOC), interviewed the eyewitnesses and survivors in the affected areas of Pakistan and neighbouring countries. Furthermore, interviewees named various inundated locations and extents in different areas. Based on these eyewitness accounts, the locations on a map and DEM were pinned up to find the inundation extent and run-up heights in the Pasni, Ormara, and Gwadar (Lodhi et al., 2021). In the follow-up studies; the Pasni was revisited to mark the inundation extents and some ambiguities in the map of Lodhi et al. (2021) were found. Based on the results of the fieldwork, our models are validated. We mapped the maximum inundation limits of the 1945 tsunami as witnessed by 1945 tsunami survivors, along with sedimentary field evidence, and simulated various wave scenarios to find the best possible fit. The 1945 wave heights at Pasni are reported differently (Table 4). This is why we used 1945 runup heights measured in the field and simulated various scenarios to estimate the waveform(s) capable of demonstrating the 1945 scenario.

## 2 Physio-geographical setting and potential tsunamis triggers

### 2.1 Tectonic settings and associated seismicity

The majority (80%) of tsunamis on the globe are caused by underwater earthquakes (Harbitz et al., 2014; Løvholt, 2017) , while the other 20% are brought on by landslides, volcanoes, and meteoric impacts in ocean water (Behrens et al., 2021). In the Arabian Sea, most of the seismic activity is attributed to the Makran Subduction Zone (MSZ). The MSZ accommodates the crustal shortening between the Arabian Plate in the south and the Eurasian Plate in the north, converging at a rate of 0.5–20 mm/year (Vernant et al., 2004; Khan et al., 2008; Jade et al., 2017). The majority of the seismicity along the eastern MSZ, south of Pasni, is attributable to the Sonne Fault (Kukowski et al., 2000). On the other hand, the high seismic velocities (4.4 km/s) suggest a brittle decollement at MSZ which is responsible for high seismicity (Smith et al., 2012). The lack of soft sediments in comparison to its western portion is what causes the brittle nature. The MSZ's thermal modelling shows its potential is comparable to the December 2004 Sumatra rupture and could trigger up to a (Mw 9.2) earthquake (Smith et al., 2012). Estimates from a different thermal modelling research suggests a potential of Mw 8.65 ± 0.26 in western Makran (Iran) and 8.75 ± 0.26 in eastern Makran, Pakistan (Khaledzadeh and Ghods, 2022). The focal mechanism of the 1945 tsunamigenic earthquake shows that it is related to a thrust event (Byrne et al., 1992). The ruptured lengths predicted by their dislocation models, body waveform inversion, and moment estimations range from 100 to 200 km with an average 7 m slip of crust along the MSZ. The Arabian Plate, along with subducting its northern edge, slides past the Indian Plate at a rate of about 3±1 mm/yr (Reilinger et al., 2006; Rodriguez et al., 2011) making it an active transform plate boundary. Right-lateral strike-slip and normal faulting cause minor earthquakes to occur in the Murray Ridge (Banghar and Syke, 1981). The largest




magnitude observed along the remaining OFZ is Mw 5.8, and the seismicity is likewise mild to moderate (Rodriguez et al.,
65  2011).

## 2.2 Gravitational mass wasting

A tsunami can be caused by an undersea landslide that moves a significant volume of water (Heidarzadeh and Satake, 2017; Röbke et al., 2018; Salmanidou et al., 2019). It has also been proposed that the two most recent tsunamis along the Makran Coast (1945 and 2013) were generated by offshore landslides triggered by earthquakes (Heidarzadeh and Satake, 2014;
Hoffmann et al., 2014). The primary explanation for the landslide source hypothesis is the tsunami's arrival delay (1-3 hours). After modelling three potential secondary sources (splay faulting, delayed ruptures, and underwater landslides), Heidarzadeh and Satake (2017) hypothesised that only a submarine landslide with a volume of around 40 km3 (see Fig. 1A) may replicate the close-range tsunami that struck Pasni and Karachi in 1945. While Yanagisawa et al. (2009) proposed that a 3-hour delay is caused by the multi-wave reflection and amplification phenomena using inverse numerical modelling.

According to Salaree and Okal, (2015), loose undersea sediments can often survive angles of 12º to 30º. However, other conditions, particularly seismic activity, can cause a landslip to occur at an angle that is much less steep. Due to a 7 km thick sedimentary sequence in the Makran Accretionary Prism (MAP) (White and Louden, 1982) and a 500 m thick pelagic drape over Owen Ridge (Party et al., 1989), the Arabian Sea has a significant potential for large offshore landslides. Numerous circular and linear slump scars may be seen on the bathymetric map and seismic lines of MAP (Kukowski et al., 2001;
Ellouz et al., 2007; Grando and McClay, 2007). The thrust faults propagated by folds have significantly increased in height and relief where the mass collapse is often observed on the frontal, south-dipping folded limb. The slope failures appear to be high but constrained by size (Mouchot et al., 2010; Platt et al., 1985). The bulk motions eventually become turbidites and settle in the deeper areas. There is a good association between the age of the Late Quaternary turbidite series (sedimentary cores) and the earthquake record (Prins et al., 2000; Bourget et al., 2010).

The southernmost Owen Ridge (see Fig. 1A) has the biggest failure area, the most significant number of landslides, and the highest estimated volume of activated material at up to 45 km$^3$ (Fournier et al., 2011; Rodriguez et al., 2013). They argue that while earthquakes are more common and mass failures are severely constrained by sedimentation rates, mass waste frequency is low due to the chaotic nature and slow sedimentation rate. The Indus Delta's landslide potential is least studied in its easternmost region. According to Milliman and Syvitski (1992), the Indus River used to supply the Arabian Sea with at
least 250 Mt/year; however, some estimates place the amount closer to 100 to 675 Mt/year (Ali and De-Boer, 2008). The Indus River has deposited 4,050–6,675 km$^3$ of sediment into the Arabian Sea since the last glaciation, and roughly half of these are deposited offshore on the shelf and in the undersea Indus Canyon (Clift and Giosan, 2014). Further research is necessary as the delta is prone to landslides due to the huge sediment buildup in the offshore and canyon zones.



## 3 Methods

### 3.1 Establishing the tsunami wave potential in the Arabian Sea

Regardless of tsunami source, we established wave potential using the tsunamite proxy and took up two cases exhibiting extreme morphodynamics discovered around the Arabian Sea (Table S1 supplements). Also, the measured runup heights of the 1945 tsunami at Pasni are assessed to measure the same potential. Both tsunamites cases are reported on the same site situated between Fins-Tiwi, Oman. The block and boulder deposits up to 40 tonnes lying at a maximum height of 10 m from the mean sea level (msl) show a landward transportation of a maximum 50 m from their intact sites. For dislodging, elevating, and transporting, an estimated wave velocity of 4.5–6.6 m/s is computed (Hoffmann et al., 2013a). The marine shells ($C^{14}$, n=4), attached to these blocks and boulders gave a wide range of age between 250±160 and 4595±185 cal. BP (Hoffmann et al., 2020). At the same site, the second tsunamite case of loose sediments fining and wedging landward with a minimum elevation of ≥17 m is dated 5840 cal. BP ($C^{14}$, n=6) (Koster et al., 2014). They also applied corrections for tectonic uplift/subsidence, absolute sea level change and estimated a 14 m wave run-up.

We simulated various scenarios and estimated that a wave with a crest height of 15 m can have a maximum 5 m/s flow velocity with a maximum flow depth of 9 m at the cliff edge (Supplements: Simulations-1 Fins-Sur, Oman). We generated the wave through earthquake parameters at MSZ (Mw 9.5, fault slip 15 m, width 200 km, depth 30 km, dip 7°). The long axis rose plot of blocks and boulders also points to the wave source of palaeocurrent (N30E) as the MSZ (Hoffmann et al., 2013a). The other tsunami sources have not yet been taken into account and therefore cannot be negated.

The 1945 tsunami at Pasni is also taken as a severe event in the known history of tsunamis in the Arabian Sea. The runup heights for this event were recorded during field work by tracing the localities pointed out by the survivors of the 1945 tsunami (Table 2) and inundation extents (see Fig. 1B). A maximum runup of 10 m is recorded at Parhag and Wadsar towns. We simulated several wave scenarios to find the best possible fit for these runups. After calibrating and validating our model with the two extreme cases discussed above, we further ran three different wave scenarios (7 m, 10 m, and 15 m) for risk analysis studies. Along with Pasni, the same wave scenarios are also projected for Gwadar for risk estimation.

### 3.2 Tsunami Modelling and simulation

Static and dynamic (hydro and/or morpo) approaches are widely used for inundation analysis to determine the flooding dynamics as flow depths and velocities along with inundation extents. We employed both approaches to better understand these dynamics and for an inter-comparative analysis both approaches as well. These approaches and models are explained separately and summarised in Table 3.

### 3.2.1 Static approach (Standing wave model)

Static inundation analysis is a quick method to demarcate the areas under different risk levels. We employed the static model developed by Berryman (2006), based on the projection of a single standing wave with equal wave height at each point on



the shoreline. The equation (i) in Table 3 calculates the tsunami height loss per cubic metre based on the value of the distance to the slope and the surface roughness. Next, the maximum inundation extent is calculated by equation (ii). Using the model builder scheme in ArcGIS 10.x, Anshori (2019) developed a toolbox to operate these equations, which reduces time and effort. The model is applied to the coastline of New Zealand by Berryman (2006), Temon, Indonesia by Prasojo et al. (2017) and Yogyakarta, Indonesia by Steinritz et al. (2021).

Finally, for both approaches, the terrain roughness is assigned using Manning's Coefficient with values ranging as 0.015 sea floor and open land, 0.03 thinly populated areas, 0.05 moderately populated areas to 0.08 densely populated areas. The coefficient's assignment is based on fieldwork, photographs, and Google Earth Satellite Images.

### 3.2.2 Hydro-morphodynamic approach (Numerical simulations)

Numerical modelling is performed using earthquake source parameters at MSZ using the Delft3D 4.04.01 software,
developed by the Dutch Institute Deltares (Deltares, 2017). The hydrodynamic programme module Delft3D-FLOW is used for simulation, which solves non-linear 2D and 3D shallow water equations for unsteady flow phenomena. The shallow water equations (SWE) of the Boussinesq approximation for incompressible fluids are derived from the three-dimensional hydrodynamic Navier-Stokes equations as represented by water and are in good approximation. The Delft3D-FLOW module performs simulation calculations of tsunami waves using the two-dimensional, depth-averaged calculation method (for
further details, please see Deltares 2012: 183–279).

We used a nested computational grid for both cities. The grid resolution for the nested models is 0.0004×0.0004 (44 m × 44 m) and 0.01 × 0.01 (~1 km × 1 km) for the regional model. A schematic nesting model is presented in Supplements: Simulations-10. In the regional model, Reinmann boundary conditions are used to minimise the reflection effect and time-series boundaries for the nested models. For tsunami propagation and inundation, the General Bathymetric Chart of the
Oceans (GEBCO) 15 arc seconds and the German Aerospace Centre's (DLR) 0.4 arcsec (~12 m) DEM are fused together. DEM is produced from bistatic X-Band interferometric SAR data using the DLR's patented TanDEM-X and TerraSAR-X. The post-data acquisition developments are manually added to the fused DEM. These changes are significant for the Gwadar as the East Bay Express Way and jetty (4 m high each) which are manually added to the DEM. There has been no significant change in the Pasni topography since 1945. A few 1945 tsunami survivors claim about the co-seismic subsidence (~2 m) at
the NE part of the city. As of 2023, this subsided topography has almost been replenished by the beach sand accretion processes. Along with the inundation analysis at Gwadar, we modelled the nearby towns of Pishukan and Surbandar. The resolution of DEM used for Gwadar city is 12 m while for Pishukan and Surbandar it is 90 m.

### 3.2.3 Vulnerability and risk analysis

A preliminary risk analysis is carried out by comparing our modelling results with the fragility curves for structural damage
developed by Koshimura et al. (2009) and Rehman and Cho (2016). These curves are developed for low-rise, lightly reinforced, non-engineered RC (see Fig. 8). The major structure types in Pasni and Gwadar are single-double story structures





with conventional masonry and lightly engineered reinforced concrete (RC). The structural and building material types in the Gwadar and Pasni are based on visual estimates carried out during the field work and need to be worked on in detail for better results. The drag force (hydrodunamic) per unit width of a structure, expressed mathematically as follows,


$$F = \frac{1}{2} C_D \rho u^2 D$$

where CD is the drag coefficient equal to 0.99 (1.0 for simplicity), $\rho$ is water density (1,000 kg-m$^{-3}$), u is the wave velocity (m-s$^{-1}$), and D the inundation/ flow depth (m). The damage probability for each scenario is calculated using hydrodynamics at the monitoring points (M1 to M7) in both study areas.

## 4 Results and interpretations

The results of both models are presented city wise and under each heading, the hydrodynamic results are followed by the static modelling results.

### 4.1 Pasni 7 m Scenario

The first wave took 20 minutes from the shore to reach its maximum height, or runup height, at Wadsar and Parhag (Supplements: Simulations-2 Pasni 7 m scenario). The waves have the potential to partly inundate Wadsar town and Main
Bazar towns (see Fig. 2A). Further northward, the inundation reaches the main road of Pasni city. The Shol, Parhag, and major portions of the main Pasni town remain unaffected by this scenario. The inundation pattern in the 7 m wave scenario is simple as compared with the 10 m and 15 m scenarios. The reflection phenomenon is also faint and could not generate higher waveforms, due to which the impact of the second wave (4 m) is significantly less. The static and dynamic modelling results best match this scenario (see Fig. 2B). To our assessment and understanding, the reflection and amplification
phenomena lose their strength as the wave scenario gets smaller. So, we expect an even better match for further smaller wave scenarios.

### 4.2 Pasni 10 m Scenario

In the 10 m wave scenario, along with the approaching wave a secondary wave is generated due to the reflection from the cliffs NE of the Pasni (Supplements: Simulations-3 Pasni 10 m scenario). These two waveforms approach Pasni Beach in 20
minutes and 45 minutes, respectively, after the earthquake (see Fig. 3A M6). Due to the large wavelength, the reflecting wave edge interacts with the approaching part of the wave, resulting in integration and amplification. The reflection source being in the northeast, the second wave pounds the city from the north. The area south-east of Wadsar (uninhabited) is most vulnerable to the extreme waves, with maximum flow depth and flow velocity of 4 m and 1.6 m/s, respectively, due to its low topography and proximity. Next, the Wadsar and the Main Bazar are the worst-affected areas. A maximum inundation
extent of 1.2 km is recorded, reaching approximately the location of Mrs. Tanoko House (see Fig. 1B). On the city centre


side, the inundating waves reach up to the huge dunes west of Pasni, sweeping the Main Bazar completely with a flow depth and flow velocity of 3 m and 1.3 m/s, respectively. Further from here, the inundation extent decreases northward. The decrease is significant, although the topography and demography are nearly the same. This is because the adjoining river (Shadi Kaur) consumes a major portion of the inundating wave, leaving behind less water budget to inundate the populated

area. However, even with this inundating pattern, the life and property losses are high. The back-wash tsunami phenomenon takes about 4 hours. A 10 m high wave ($\lambda/2= 25$km) is in closest agreement with the field observations of the 1945 tsunami. The Shol and Parhag areas in both models (static and dynamic) remained unaffected by the inundating wave. The Wadsar and north of Main Pasni Town (M6) experience an inundation depth of maximum 3 m and 2 m, respectively, which is somehow close to that of the dynamic model. However, the inundation extents on both sites are underestimated at 400 m and

450 m, respectively, in comparison to the dynamic model. Moreover, the results of Main Bazar and surrounding areas are severely compromised by the static model, where the inundation extent is underestimated by 700 m (see Fig. 3B). These lower estimates keep the Main Bazar uninundated. As already mentioned in the limitations section, the lower estimations are due to the reflection phenomenon, which is not inherited by the static model.

### 4.3 Pasni 15 m Scenario

In this scenario, a huge pile of water results from reflection, integration, and amplification which inundates deep into the Pasni and surrounding areas (Supplements: Simulations-4 Pasni 15 m scenario). The refection phenomenon has extremely devastating consequences as two gigantic reflecting waves completely drown the whole populated land. The whole Pasni, Wadsar and Parhag experiences a flow depth of 6-8 m, except Shol, where it is 3 m (see Fig. 4A). The inundation impact in terms of flow depth and extents doubles as compared to that in 10 m scenario, especially the later with flow depths of 7 m at

the Wadsar monitoring point (M3). Based on the above observations, it is interpreted that the number of waves and the time delay at Pasni have a relationship with the wavelength of the approaching wave. The time delay proxy may be used for the inverse calculations to estimate waveforms during the 1945 tsunami, but the information on the time delay of the 1945 Tsunami is not adequate to perform the test. The depth average flow velocities show anomalous observations, as near the Pasni (M2), the peak value observed is only 2 m/s against a flow depth of 8 m. We interpreted this as due to the energy loss

caused by the reflection, integration, and amplification phenomena.

The static model severely underestimated the results as compared to those of the dynamic model. The static model could not inundate the Shol, Parhag, and half of the main Pasni town; however, these areas experienced 3 m, 6 m, and 8 m flow depths in dynamic modelling (see Fig. 4B). The reason is the same as discussed before (reflection and integration of waves), and in this scenario, the phenomenon is further amplified. Amplification exposes the coast to large volumes of water as compared

to a static or single-wave impact.





### 4.4 Gwadar 7 m Scenario

The 7 m wave scenario affects Gwadar and Surbandar; however, the Pishukan remains unaffected. At Gwadar, the waves inundate both bays; on the eastern bay, Mohalla and the adjoining seaport are affected with flow depth and flow velocity of 3 m and 1.6 m/s, respectively (see Fig. 5A M1). This is the same area that underwent severe damage in the 1945 tsunami

(Kakar et al., 2014). The morphology of Gwadar city has changed much from what it was in 1945 due to the construction of raised road highways that are averagely 4 m (from the high tide line) high on both sides of the bay bordering the city. These bordering structures act as defence structures against tsunamis and storm related waves. The 1945 tsunami height at Gwadar was not exactly recorded or reported. According to our analysis based on the survivor's interviews and dynamic modelling results, the tsunami wave height was around 6 m to 7m. Only the eastern side facing the tsunami source was affected. As the

inundation dynamics change with source location and we projected the waves from the whole MSZ, the inundation occurs at the New Town also by flow depth and flow velocity of 3.6 m and 1.3 m/s, respectively.

The defence structures temporarily dammed the city, due to which the backwash phenomenon took a considerably large amount of time. Also, a poor and low-capacity drainage network further retards the process. The Mohalla, spread over a 4.5 km2 area, remained under water at a depth of 2 m even after 4 hours (see Fig. 5A M1). At New Town (~4 km2), backwash

further aggravates, with a water depth of 2.5 m for the complete period of simulation (see Fig. 5A M2).

Reflection, integration, and amplification are common devastating phenomena observed in Pasni and Gwadar. In the case of the 7 m scenario, three reflective waves are generated in addition to the approaching wave (see Fig. 5A M1). Of these three, two are 7 m high and one <0.4 m high. The reflective waves, while propagating southward, strike the Gwadar-Hammer-Head structure (highlands) and find their way to the Mohalla Band area (Supplements: Simulations-6 Gwadar 7 m scenario). Each

wave exacerbates the effect of inundation. This effect is conspicuous at Mohalla Band monitoring point (M1), where the flow depths and flow velocities increase gradually from 1 m and 0.5 m/s (wave-1) to 3 m and 1.0 m/s (wave-3). More or less similar inundation dynamics occur at the New Town locality on the western Gwadar Bay. The proposed future industrial site is also vulnerable to this wave scenario. The Surbandar locality narrowly escaped the tsunami inundation. Though the waves overtop or overflow the beachbanks, following the low-lying topography, the tsunami torrent bypasses the populated area

and drains back into the sea. The Pishukan, due to its high topography, stays safe in this scenario. The boats at the docking site or related installments may suffer minor damage.

The static model results are only comparable to those of the first dynamic wave (see Fig. 5B); however, the simulation shows that the worst comes with reflection and amplification phenomena. For this reason, the static model overestimates the beach proximity and underestimates the inundation extents.

### 245 4.5 Gwadar 10 m Scenario

The inundation pattern of the 15 m scenario is almost identical to that of the 7 m scenario, but the inundating dynamics (flow, depth, and extent) are more than twice as large as those in the 7 m scenario. Almost complete Gwadar-Hammer-Head-





Neck (Gwadar city) and Surbandar experience inundation, and Pishukan remains un-inundated again (see Fig. 6A). The Mohalla Band and New Town receive three waves, which gradually increase the flow depth from 3 m to 5 m and 5 m to 7 m, respectively. At the New Town, the approaching-reflective wave interval is very short, due to which both wave forms superimpose over one another and make a plateau-shaped flow depth graph. The inundating extents from both sides of the Gwadar Bays overlap to completely submerge the town (Supplements: Simulations-7 Gwadar 10 m scenario). A maximum inundation extent of 4.5 km is observed across the industrial area (M3), facilitated by a stream in its northern vicinity. It is observed that maximum flow velocity is experienced when a site is directly facing the approaching wave. Also, much of the wave energy is lost during the reflection phenomenon. These two observations are based on a comparative analysis of flow velocities between approaching and reflecting waves. For instance, at the Mohalla Band, an approaching wave has a maximum flow depth of 3 m and a flow velocity of 2 m/s. Further, the two reflected waves increase the flow depths to 5 m, but flow velocities decrease to <0.5 m/s. The lateral two piled up huge amounts of water at the coast, but energy (velocity) decreased around threefold. Surbandar is the only locality that directly faces the approaching wave and experiences the highest flow velocity of 4 m/s with a flow depth of 6 m. The static model did not produce adequate and satisfactory results in this scenario. It overestimates the flow depths approximately by a factor 2 at the beach proximity and underestimates the inundation extents as it lacks parameters to estimate the reflection input (see Fig. 6B).

**4.6 Gwadar 15 m Scenario**

In the 15 m wave scenario, the reflection and amplification phenomena further gain momentum. For instance, the peak flow depth at Mohalla Band (M1) in a 10 m scenario and a 15 m scenario is 5 m and 12 m, respectively. Here, the flow depth in the latter case is increased by a factor of 2 in relation to the approaching wave height and wavelength. Also, with a higher wave scenario, the time interval between the reflecting waves decreases, resulting in a quick overlap of major wave portions, and only small wave crests are observed in the flow depth graph trajectories. At the Mohallah Band, New Town, Industrial, and Surbandar localities, the immense inundation takes place for about 25 minutes and then starts the backwash (see Fig. 7A). As already discussed, these two localities are embanked by the raised road highways. The entrapped or dammed water only finds its way through the sewerage or drainage network. This is why the first two localities remain more than 2 m under water even after hours in all scenarios. The Gwadar-Hammer-Head receives a weekly wave reflection after four hours, which has minor effects on the flow depths and flow velocities (see Fig. 7A). In a 15 m scenario that inundates the whole Surbandar, the maximum flow depth is 11 m with a very high flow velocity of 6 m/s. In this scenario, the flow depth doubles as compared to the 10 m wave scenario. The scenario will be terrible as an approaching wave overtops the beach cliffs; the northward downstream rushing water will completely scrape off the town (Supplements: Simulations-8 Gwadar 15 m scenario).

As it overtopped the ridge, the Surbandar community may not visually notice the approaching wave before impact, due to which the earlier unwarned community only has less than a minute to climb up the higher grounds. Consequently, the casualty rate may be much higher here. The Pishukan town could not evade this scenario and receive overtopped waves from



the southern beach cliffs as well as those of Surbandar. It experiences a maximum flow depth and wave velocity of 3 m and 2 m/s, respectively. With the approaching wave visually hidden behind the cliffs as in Surbandar, the earlier unwarned community has a maximum of 4 minutes to find the safe sites. Due to the high impact of reflection, integration, and amplification processes, we did not carry out the comparative analysis between static and dynamic modelling. However, the static model highly underestimates the scenario (see Fig. 7B).

285

## 5. A preliminary risk analysis

The damage probability for each scenario is calculated using hydrodynamics at the monitoring points (M1 to M7) in both cities. Therefore, from these monitoring points, damage intensity will gradually increase shoreward. The cases of Surbandar and Pishukan in Gwadar are exceptions discussed later. At Pasni, the 7-metre scenario minorly affects the Main Bazar and surrounding areas with an approximate damage probability of 0.3. The M1 and M2 have a higher damage probability, and these areas are barren and unpopulated yet. The scenario simulation shows that it will affect Wadsar town partially, and the inundation has not reached the monitoring point. The 10-metre scenario will affect Wadsar, Main Bazar, and the areas around the M6 with a maximum damage probability of 0.6, 0.4, and 0.1, respectively (see Fig. 8). Again, the worst-hit areas are M1 and M2, which are uninhabited yet. As already mentioned, this scenario best fits the 1945 tsunami scenario, and interpretation is made solely based on inundation extent. The damage probabilities would have been higher then due to the single-story houses being made of wood and mud only. The 15-metre scenario spares no area in either of the cities. The Shol area is geomorphologically protected and farthest from the shoreline. However, the damage probability here is greater than 0.4. The worst affected areas are Wadsar and Parhag, with a damage probability of 0.9. The Main Bazar and area around M6 are also at high risk, with damage probabilities ranging from 0.7 to 0.9 (see Fig. 8). If the areas around M1 and M2 are developed with the current construction type, then complete damage is expected as the probability exceeds 0.9.

At Gwadar, a 7 m wave scenario is capable of affecting the Mohallah Band, New Town, and the proposed Industrial Zone. Here, maximum destruction is expected at Mohallah Band, with a damage probability of 0.4. Also, the adjacent ship docking and trade port facilities will undergo considerable damage. The simulation shows that at the Surbandar, the overtop tsunami torrent follows the low-lying area and does not reach the Measuring Point (M4); however, a few houses in between may be at risk. In the 10 m scenario, all four localities will be seriously disrupted. The former three (Mohallah Band, New Town, and Industrial Zone) have a damage probability of around 0.9, while Surbandar has a probability of 1.0. The 100% probability of damage at the Surbandar is due to two factors: first, the perpendicular wave impact at the beach, and second, the downslope high flow velocity (4 m/s), which increases the hydrodynamic force (>40 kNm). The 15 m scenario completely disrupts the Gwadar area. 100% damage at all sites except 0.6 at Pishukan. The approaching wave overtops the Pishukan beach cliffs, and further downstream, the Pishukan lies downstream, receiving a high-velocity tsunami as happened at the Surbandar (explained above). Like Pasni, there had been houses made up of mud and wood in 1945, which could hardly withstand even a 2 m flow depth. According to our field observation, highly substandard concrete quality is being





used. In this regard, beach sand is being widely used for all purpose constructions, and it may be one of the major ingredients hampering the strength of mortar or concrete. The beach sand grains at Pasni and Gwadar are very well rounded and

polished, which offers minimum adhesiveness to the cement epoxy. Also, the level of salt content on the beach sand is very high, resulting in post-construction crystal growth. The crystal growth exerts pressure on the structure, resulting in crack development. However, this preliminary interpretation is tentative and should be investigated in detail.

Rehman and Cho (2016) used only flow depth to calculate the damage probabilities and complete damage exceedance probability for a single-story conventional masonry and concrete coastal building. Their results conclude that under 2 m, 4

m, and 7 m flow depths, the damage probabilities are about 0.2, 0.6, and 0.8, respectively. This model overestimates the damage probabilities and also lacks the flow velocity component in the damage calculation, reducing the reliability of the model.

## 6. Discussion and Conclusions

To calibrate a tsunami numerical model with reference to field observations and readings, it is necessary to simulate a variety

of potential tsunami wave scenarios. It is to be performed on a DEM that precisely reflects the terrain conditions and hydrodynamics of the period. Due to the complex interplay of hydro-morphodynamics, each tsunami can be explained mathematically by various wave theories (Tadepalli and Synolakis, 1994; Carrier et al., 2003; Madsen and Schaeffer, 2010). Each theory has its own waveform, intensity, wavelength, duration, and geographical reach, which makes the calibration complicated. As we calibrated our model with the 1945 tsunami scenario, there are certain aspects that match well with the

results, while some aspects of the model could not be fully satisfied. However, the analysis of different wave scenarios enables us to understand the inundation pattern and identify the areas at various risk levels in any potential case. The inundation extents and flow depths match well, but the aspects of the three inundating waves (N-waves) as reported by 1945 tsunami survivors are not yet fully understood. Regarding the number of waves, we presented our analysis in the results section. Briefly, it is the reflection phenomenon responsible for the generation of second and/or third waves. Another

concern is the reported arrival time delay of one to two hours at the Pasni. Besides the time delay's reporters, three elderly citizens also reported it to be about half an hour (Table 4). Moreover, there is a reasonable match between the simulation time and the actual tsunami arrival time at the Karachi and Mumbai Tide Gauge Records of 1945. These two instrumental records did not account for the time delays and a bigger wave later, due to which there appears to be an exaggeration in the facts of the time delay.

The simulation shows that reflection phenomena gain strength with an increase in wave height and wavelength. The sensitivity analysis shows that the former is a function of the slip component, while the latter is a function of fault zone thickness. The reflections of large waves integrate with the proceeding wave to amplify it, resulting in the generation of secondary waves. These secondary waves, upon collision, largely lose energy at first but pile up huge amounts of water near the shores. The former one reduces the flow speed, but the latter one, on the other hand, results in higher flow depths. This




phenomenon is prominent at the Mohalla Band and the New Town in Gwadar. From the above discussion, we emphasise that the reflection effect must be considered for inundation analysis, especially for the areas lying oblique to the wave propagation and the shadowed areas (as Pasni, Gwadar) should not be considered relatively safe. The numerical model testifies to a general understanding that flow velocity on land is maximum when the impacted coast lies perpendicular to the direction of wave propagation. In such cases, the approaching wave transfers maximum hydrodynamic energy, as observed

at Surbandar. Eventually, the discussed effects will increase the vulnerability and risk factors for a coastal community. The static model is suitable only for a single wave analysis and for the coastlines directly facing the approaching wave. Perpendicular orientation minimises the reflection effect and gives close results to those of a dynamic model.

  In tsunami hazard assessment, wave height is considered a benchmark, but the wavelength is mostly ignored. We assessed that for risk assessment, along with wave height, the wavelength must be taken into consideration. For instance, we tested

two scenarios with the same wave height of 5 m but different wavelengths ($\lambda/2$) of 15 km and 3 km pounding the Pasni coast (Supplements: Simulations-9). The wave with larger wavelength did not inundate the land and dispersed quickly; however, the larger wavelength considerably inundated with runup height and inundation extent of 2 m and 0.7 km, respectively. Due to the shorter wavelength, the quantity of displaced water is less, which disperses quickly and loses magnitude with propagation, making these hazards local only. Landslide-related tsunamis have very short wavelengths as compared with

earthquake tsunamis.

  The surface roughness factor, or coefficient of friction, is an important factor in controlling the inundation depth and flow velocities. We used values matching present ground conditions useful in understanding the risk factor at present. The results can be improved further by precising the surface roughness, bathymetric resolution, and decreasing the computational grid size. The bare ground elevation model represents a digital topography with no building infrastructure, plants, poles, etc. It is

formally known as the digital terrain model (DTM). For inundation analysis, the deflection of DEM from DTM hampers the results in two aspects: i) the porous infrastructure, such as houses, plants, poles, etc., does not account for water balance adjustments. These structures act as solid blocks, occupying the volume and deflecting the inundating water further ahead, resulting in higher inundation extents. ii) In the case of a low-resolution DEM that cannot resolve the streets or passages results in water damming. The damming results in higher flow depths downstream than actual, and it also affects the

backflow computations.

  Due to its proximity to MSZ, Makran Coast is unable to fully benefit from the Indian Ocean Tsunami Warning and Mitigation System (IOTWS), a far-field early warning system established in the Arabian Sea. This method is beneficial for the south-western Indian coast, including Mumbai city, since far-field warning mechanism can operate for it. According to numerical simulations of the 1945 tsunami, the first wave arrives at the Pasni, Karachi, and nearest Deep-Ocean Assessment

and Reporting of Tsunami Stations (DART) in around 15, 80, and 45 minutes, respectively. Heidarzadeh et al. (2007) calculated these times as 15, 100, and 60 minutes, respectively. From these studies, there is a reaction time of around 45–60 minutes for alert issuance, warning dissemination, and evacuation. So Pasni, Gwadar and the proximal Makran Coast are likely to receive the first wave in 15 minutes and surpass the IOTWS alert issuance.



Further, once the wave is near the beach, the visual estimates to alert the community nearby beaches is extremely small. It
takes around 15 minutes to inundate from shoreline to maximum height (runup height). Here, we emphasise the need for
vertical evacuation structures and the installation of alarms at the beach, which could be activated locally at the earliest. The
reaction time is extremely short for Surbandar (<3 minutes) and Pishukan (<5 minutes). As already explained, these towns
will be inundated by the overtopping wave(s) rushing over the towns situated downstream. Approaching waves, being
visually hidden by the cliffs, may go unnoticed, leaving almost no time to rush to higher ground.

The Pakistan Meteorological Department (PMD) is the organization responsible for keeping track of coastal hazards and
providing notifications and alert issuance. A well-established cyclone early warning system was built by PMD in the 1960s,
and a near-field tsunami early warning system in 2010. The high number of fatalities and property losses from the cyclones
in 1998, 1999, 2007, 2010, and 2021 demonstrate significant susceptibility and represent dismal levels in terms of resilience
and preparedness. Furthermore, the impact of a tsunami at these levels would be far harsher.

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

**Figures and tables**



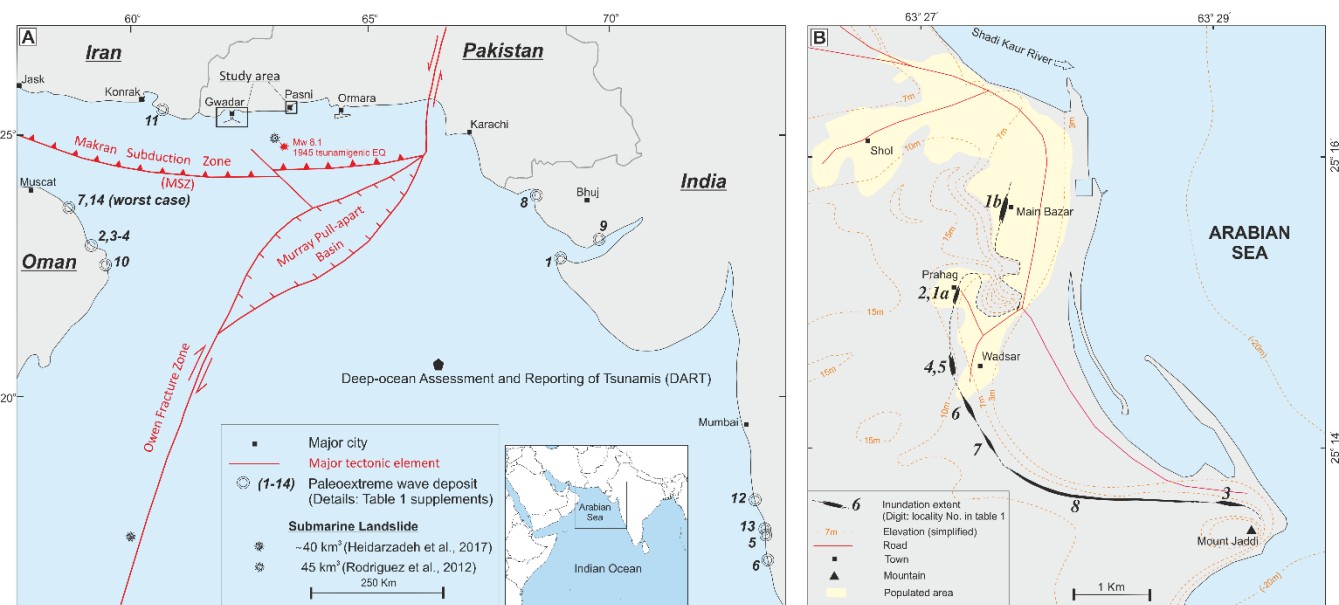


**Figure 1: A Location and tectonic map of the study area. The circles with numbers show the reported palaeoextreme wave deposits; details are in Table S1 in the supplements. B Map showing inundation extents during the 1945 tsunami in the Pasni area. The mapping is based on the interviews and the field observations given in Table 2.**

**Table 2: Table showing the compilation of inundation facts at Pasni from the interviews of 1945 survivors and the field**
**observations.**

| Interviewee | Age (years as of 1945) | Brief description | Run-up height | Inund. extent | Locality No. in Figure 1B |
|---|---|---|---|---|---|
| Tanakko Bibi | 11 | a) "The 1945 tsunami came up until here, where we are sitting right now (her house). You can still see the seashells (of the 1945 tsunami) scattered around here."" | 9 m | 1.6 km | 1a |
|  |  | b) "Much of the casualties happened in the Main Bazar locality" | 7 m | 1.3 km | 1b |
| Rabia Bibi | 4-5 | "Water came to Tanakko Bibi's house" | 9 m | 1.6 km | 2 |
| Ganj Baksh | 14-15 | "Tsunami destroyed houses, boats, and debris nearly as far inland as Parhag. Many houses and boats were stranded beside Jaddi Hill." | 8 m | 360 m to 1.52 km | 3 |
| Khudai Dost | 10-15 | "Part of Wadsar was drowned" |  |  | 4 |
| Qadir Baksh | 15 | "Tasunmi water came to Daragah (Shrine) and then receded back. There were already scattered bivalve shells on the inundated land, but the concentration had increased after the 1945 tsunami" | 10 m | 1.6 km | 5 |
| Field work |  | The above reported localities were identified in the field. We traced a curvilinear deposit of broken bivalve (shell bed) connecting these above-mentioned localities and interpreted it to be reworked deposits from the 1945 tsunami (tsunamites). | 8 m | 0.3 – 1.8 km | 6,7,8 |





**Table 3:** Summarised methodology with the respective basic components used in each model.

| Approach | Model | Equation/s | Reference |
|---|---|---|---|
| Static | **Standing wave**<br>• mass conservation<br>• surface roughness | $Head_{loss} = \left(\frac{167n^2}{Y_0^{\frac{1}{3}}}\right) + 5\sin \sin s_0$ ------(i)<br><br>$Inundation_{max} = \frac{Y_0}{Head_{loss}}10$ -------(ii)<br>n: roughness coefficient<br>$s_o$ : Topographic slope (radian)<br>$Y_o$: Initial wave height on the shoreline | (Berryman, 2005)<br>(Prasojo et al., 2017)<br>(Juniansah et al., 2018)<br>(Steinritz et al., 2021) |
| Hydrodynamic | **Delft3D**<br>• mass conservation<br>• momentum eq.<br>• surface roughness<br>• simulations | • *Hydrodynamic equations*<br>• *Continuity equation*<br>• *Momentum equations in horizontal direction*<br>• *Vertical velocities*<br>• *Hydrostatic pressure assumption*<br>• *Coriolis force and Reynolds stresses equations*<br><br>* For details, referred to (Deltares, 2017), pp. 175–181) | (Deltares, 2017)<br>(Gelfenbaum et al., 2007)<br>(Chacón-Barrantes et al., 2013)<br>(Röbke et al., 2016) (Röbke et al., 2018)<br>(Van Ormondt et al., 2020)<br>(Sujatmiko et al., 2023) |


**Figure 2 (Scenario 7 m): A** Hydrodynamic modelling results of maximum inundation limits and depths at Pasni and surrounding areas (Supplements: Simulations-2 Pasni 7 m scenario). The graphs show the flow depth and flow magnitude at each monitoring point (M1 to M7). **B** Static modelling results showing flow depths and inundation extents.


**Figure 3 (Scenario 10 m): A Hydrodynamic modelling results of maximum inundation limits and depths at Pasni and surrounding areas (Supplements: Simulations-3 Pasni 10 m scenario). The graphs show the flow depth and flow magnitude at each monitoring point (M1 to M7). B Static modelling results show flow depths and inundation extents.**


**Figure 4 (Scenario 15 m): A Hydrodynamic modelling results of maximum inundation limits and depths at Pasni and surrounding areas (Supplements: Simulations-4 Pasni 15 m scenario). The graphs show the flow depth and flow magnitude at each monitoring point (M1 to M7). B Static modelling results showing flow depths and inundation extents.**
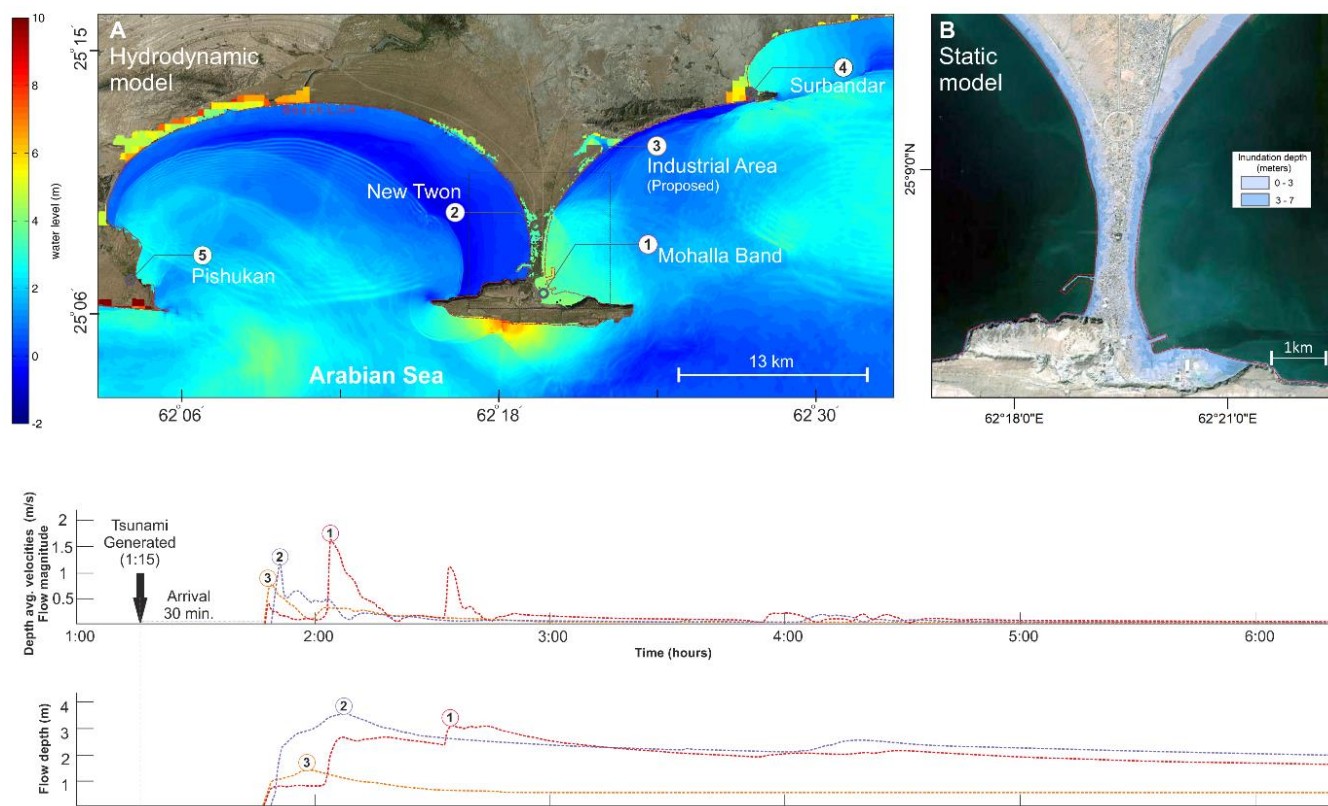


**Figure 5 (Scenario 7 m): A Hydrodynamic modelling results of maximum inundation limits and depths at Pasni and surrounding areas (Supplements: Simulations-6 Gwadar 7 m scenario). The graphs show the flow depth and flow magnitude at each monitoring point (M1 to M7). B Static modelling results showing flow depths and inundation extents.**

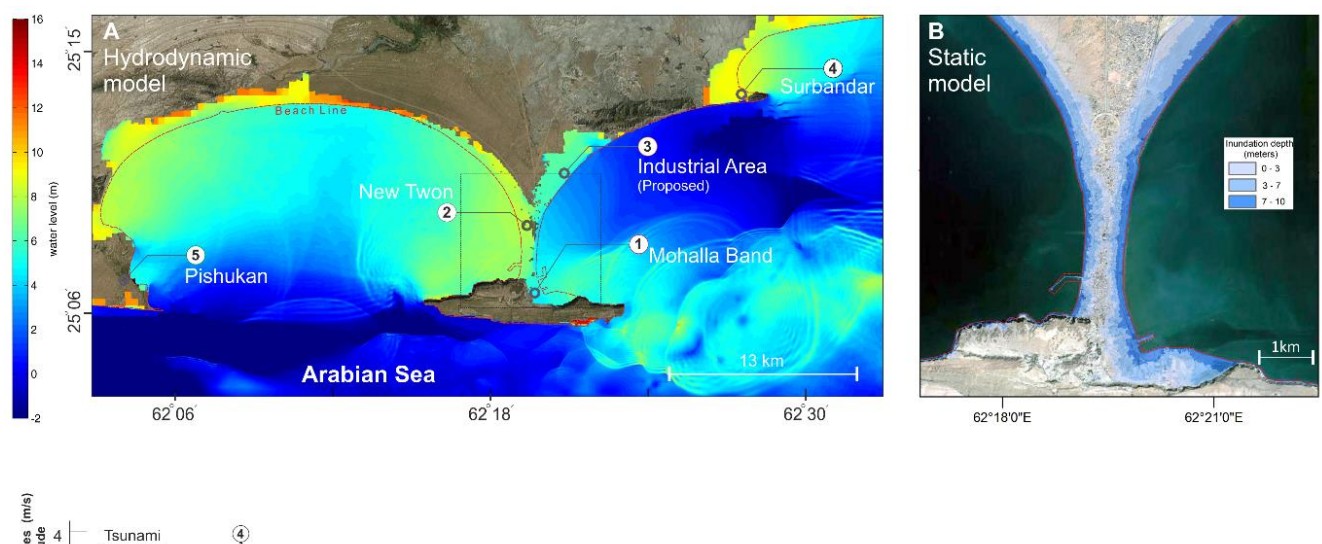

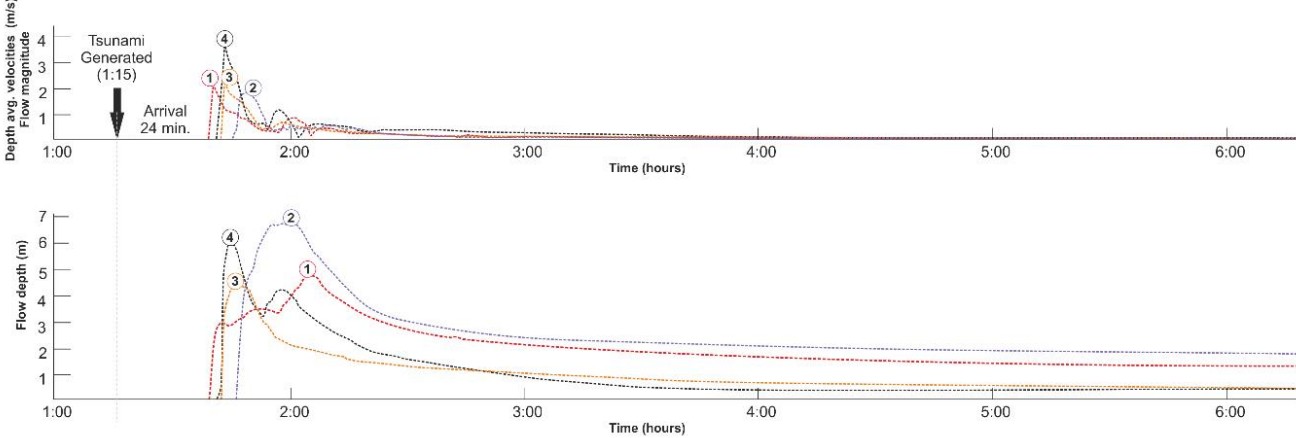


**Figure 6 (Scenario 10 m): A Hydrodynamic modelling results of maximum inundation limits and depths at Pasni and surrounding areas (Supplements: Simulations-7 Gwadar 10 m scenario). The graphs show the flow depth and flow magnitude at each monitoring point (M1 to M7). B Static modelling results showing only flow depths and inundation extents.**

**Figure 7 (Scenario 15 m): A-Hydrodynamic modelling results of maximum inundation limits and depths at Pasni and surrounding areas (Supplements: Simulations-8 Gwadar 15 m scenario). The graphs show the flow depth and flow magnitude at each monitoring point (M1 to M7). B-Static modelling results showing flow depths and inundation extents.**





Figure 8: A Probability structural damage plotted against the hydrodynamic force at each monitoring point. Top row: Pasni, Bottom row: Gwadar. The results plotted on fragility curves developed by (Koshimura et al., 2009).

Table 4: Table showing compilation of reporting wave heights and times at Pasni, Karachi and Mumbai. The line-graphs below compare the observed arrival times with the modelled one at Karachi and Mumbai (Revised after Heidarzadeh and Satake, (2014b). The references not in the text (Heidarzadeh et al., 2008, 2009), (Jaiswal et al., 2009)

| Reference | Observed Max. Wave height (location) | Observed arrival time of wave/s (1st, 2nd, 3rd) | Arrival time/delay |
|---|---|---|---|
| (Pendse, 1946) | 12-15 m (Pasni) | 04:00, 07:15, -- | "At about 4:00 IST a wave was noticed but it did not come inland. At about 7:15 IST another wave swept over the town and caused widespread havoc. The height of this wave has been estimated variously from 40 ft to 60 ft (~12 m to 18 m)" |
| | 1.4 m (Karachi) | 05:30, 07:00, 07:50 | last wave was highest with wave height of 1.4 m |
| | 1.9 m (Mumbai) | 08:15, --, -- | |
| (Gates et al., | 7-10 m (Pasni) | 04:00, 04:30, 5:00 | |



| | | | |
|---|---|---|---|
| 1977) | | | |
| (Page et al., 1979) | 7-10 m (Pasni) | 04:30, 05:00, -- | "Local inhabitants report that three tsunamis hit the Makran Coast 1.5 to 2 hours (4:30, 5:00 IST). Field investigation showed that there was no uplift of Pasni resulting from the earthquake, but the Ormara area rose about 2 m" |
| (Neetu et al., 2011) | 0.28 m (Karachi) | 05:00, --, -- | Karachi Tide Gauge (1945), First wave arrival 5:00 IST |
| | 0.35 m (Mumbai) | 08:09, --, -- | Mumbai Tide Gauge (1945), First wave arrival 08:09 IST |
| (Kakar et al., (2014) | ≥7 m (Pasni) | 04:00, --, -- | Three waves with interval of few minutes arrived ~30 minutes after the earthquake, Interviewees: i)-Khuadi Dost, ii)-Sakhi iii)-Daad, Madni |

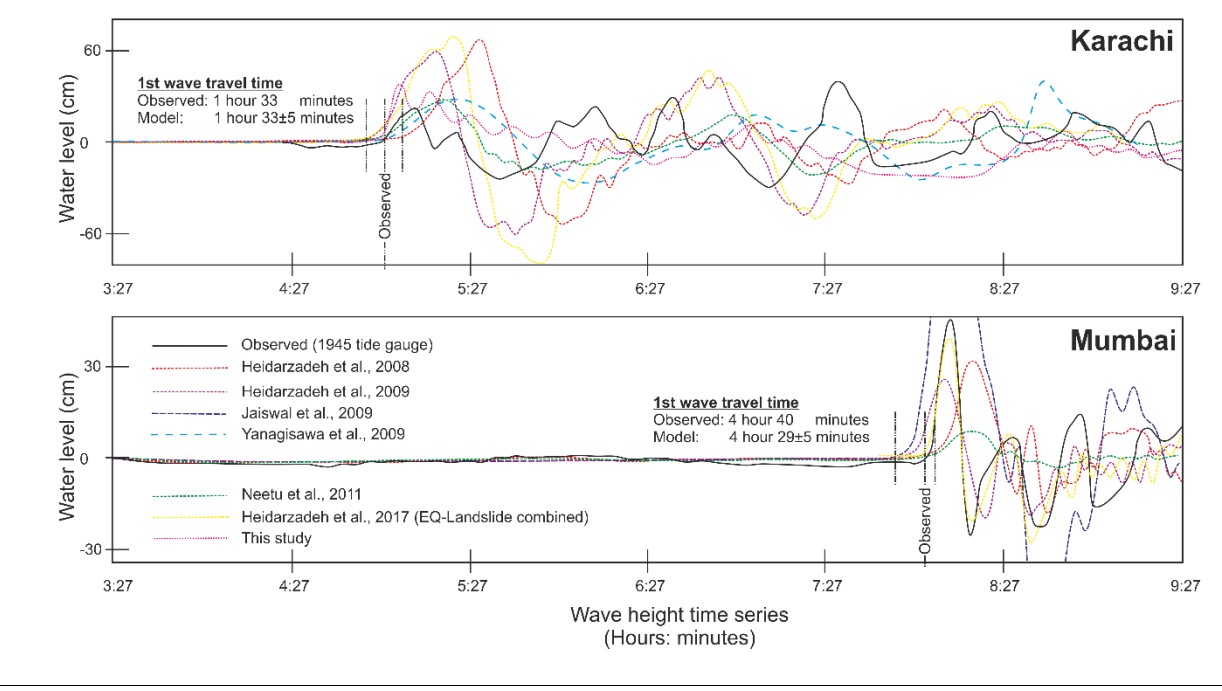


