# Peer review of "Tsunami inundation and vulnerability analysis at the Makran Coast, Pakistan"

_Natural Hazards and Earth System Sciences, 2023_

## Author Response (AR1)

**Dear Editor, Dear Anonymous Referee #2,**

We thank you for your valuable time, effort, and useful comments on our manuscript. Each comment and suggestion are addressed. detailed below, please.

=================================================================

**COMMENTS and REPLIES (UPDATED NAMUSCRIPT)**

**GENERAL COMMENTS**

1. **This application doesn't sound like a comprehensive risk analysis, while it could be more considered as an enhanced vulnerability assessment, with some elements of risk (percentage of damages on buildings).**

Title and manuscript updated considering as an enhanced vulnerability assessment instead of risk assessment as advised.

2. **Descriptions are too approximate, leaving too many uncertainties to the reader who isn't an expert on the area or of the topic, and taking too much for granted. There are many references to toponyms that are not reported in the maps; some concepts are anticipated before description (e.g.: the reference to static and dynamic modelling in the abstract; citing reflection phenomena in Pasni-7m scenario while they are observed only in the following results); the computing techniques and the scenario selection and definition (see point below) are described too superficially. This makes the reading difficult and the whole reasoning difficult to follow.**

The entire manuscript has undergone meticulous revision to enhance clarity and comprehensibility for the reader. Several significant changes have been made throughout the document. These major changes are as follows:

- Abstract is revised.
- Section "2.2 Gravitational mass wasting" is removed as paper deals with earthquake tsunami sources only (as advised).
- Section "3.2 Source Type and Parameters" is added (as advised) including table-1 showing fault parameters for each scenario with respective initial wave height figure. Location of sources (faults) is added in Figure-1.
- The static approach to model inundation is removed as advised and now only incorporating the dynamic modelling. It's not the specific modelling demand or requirement and secondly to avoid reader's mind clustering.
- We removed the discussion of earthquake magnitude as it left cross-questions and confusions, now we are dealing with the earthquake parameters only.
- Computing techniques are explained further, and a flowchart is added along with Table-2 showing sequential order of various processes used for hazard and vulnerability assessment.

- All tsunami scenarios are revised and rewritten in specific manner under one heading for each area i.e. Gwadar and Pasni.

- The references to toponyms are reduced and explained in more detailed. The figures are labelled accordingly. The figures layout is revised. For Pasni, the Figures-2, 3, 4 are combined into one single Figure-2 giving more comprehension and comparability. Similarly, for Gwadar, the Figures-5,6,7 are combined into one i.e. Figure-3.

- We removed the correlation of our inundation results with the tsunami case of 1945 due to time lapse and related morphic changes hampering the result's correlation. In this regard, the Figure-1 (Panel-B) and Table-2 are removed also.

- Table-4 is added as supplement showing the timeline of events of three tsunami cases in the Indian Ocean to evaluate the warning alerts performance of the system and the operators. It involves one near-field tsunami case and two far-field cases.

3. **The selected scenarios are questionable: the assumption of 7m, 10m and 15m cases should be justified better.**

   **Which sources are used to produce them? Only 15m scenario is briefly accounted for. Is it possible to associate each earthquake/tsunami scenario with a return period, for example, or at least justify them in terms of past events?**

We gave the justification for the choice of scenario in section "3.2 Source Type and Parameters" with addition of recurrence interval referring to our one of previous research paper.

The present study is a continuity of a hazard and vulnerability analysis project along the Makran Coast, Pakistan. We had concluded the hazard assessment part though a multiproxy analysis and published (Haider et al., 2023). The current candidate-publication represents a progressive step towards advancing the vulnerability assessment within the same study area. In this manuscript, we are adding these findings briefly referring to Haider et al. (2023).

4. **Is it correct to use 1945 pieces of evidence as a proxy for simulations that adopt present morphology, especially for roughness coefficients governing inland flooding? Doesn't this affect the simulations?**

We removed the 1945 Tsunami case form this candidate publication. Therefore, this research is based on the present morphology beneficial more for today's community.

**SPECIFIC COMMENTS**

**Abstract**

- **Line 12: citing "dynamic and static approaches" before describing them can be misleading, I would change the sentence.**
- **Line 13: not all readers could have in mind the local geography: where is located the Arabian Sea with respect to the investigation site?**

We deleted the static model and its findings from the paper. The manuscript only deals with the dynamic model now. Both comments are addressed together, and initial part of the abstract is revised also as advised.

- **Line 20: after the point, it is unclear what "It" refers to.**

We improved and replaced the pronoun "it" with its proper reference "reflection-amplification phenomenon"

**Introduction**

- **Lines 43-45: the two sentences are very confusing, it is hard to understand their meaning, rephrase them.**

   We addressed as advised.

**Physio-geographical setting and potential tsunamis triggers**

- **Is paragraph 2.2 necessary? Of course, the source type is interesting and deserves attention, but in the paper it isn't treated nor cited anymore.**

We removed the paragraph 2.2 as advised. We added "3.2 Source Type and Parameters" and moved the section under methodology heading.

**Methods**

- **Line 108: magnitude 9.5 is extreme, how can the authors justify it? In the previous section, they stated that the maximum expected Mw was 9.2, here they use 9.5. Why?**

   **It seems from the text that the 15m wave scenario is generated by the Mw=9.5 source: what about the other scenarios (7m and 10m)? Did the author obtain them through other seismic sources in the MSZ, presumably smaller?**

After critical comments from both reviewers, we are getting away from this discussion as this comparison is not the focus of the study. The relationship between earthquake magnitude and associated parameters is inherently complex, primarily due to the intricate nature of geological processes. This complexity necessitates comprehensive literature review and data collection-analysis and a deep understanding of geological phenomena. So, in the manuscript, instead of dealing with the size of earthquake and associated tsunami, we are dealing with the earthquake parameters only to stay focused on the scope of the study.

The "3.2 Source Type and Parameters" addresses the second part of the question/ comment.

**Results and interpretations**

- **Line 168: the runup usually refers to the maximum elevation reached by the water flooding, so "runup height" doesn't sound correct.**

We want to refer "maximum inundation limit", so we replaced accordingly.

- **Line 168: according to Fig. 2A, neither Wadsar nor Parhag are reached by the water.**

The 7-meter scenario partially floods Wadsar locality, affecting approximately 30% of its southern area (visual estimates). However, it's important to note that (here) the monitoring point (M3) within the town remains un-flooded. Despite the impression that Wadasar is completely inundated, this is not the case. For visual understanding, we refer to the simulation file provided in the supplementary section of the manuscript.

Moving forward, the toponyms and locality names are added to text in an improved way to track them on the figures/maps.

- **Line 170: the figure does not report the position of Pasni, that can be inferred from Fig. 1. Placing a label with position of the town can facilitate the reader.**

We labelled the "Pasni"; the main locality of the area in all the figures as advised.

- **Lines 172-173: reflection does not occur in this scenario, and commenting on it here sounds incomprehensible.**

We removed the sentence as advised.

- **Lines 178-179: the cliffs NE of Pasni are not visible in the figure, so they should be indicated.**

We indicated and labelled the cliff's location and extent in Figure-1 as advised.

- **Lines 179-180: Fig 3B, flow depth graph, doesn't seem to show two waveforms.**

The intense reflection phenomenon and reflection timings are such that, the whole tsunami wave package appears to be a single unit. However, a closer look at the flow depth graph shows small humps (wave troughs) within the big tsunami waveform which mildly shows impact of multiple waves.

- **Line 216: where is Gwadar on the map?**

We labelled "Gwadar" in the map in the Figure-3.

- **Line 233: again, Gwadar-hammer-head toponym is not reported in the figure.**

We labelled the "Gwadar-Hammer-Head" in the Figure-3.

- **Line 290: what does damage probability mean? The methodology should be explained a bit, at least. Moreover, in the picture, Main Bazar seems affected by a probability close to 0, not certainly 0.3.**
    - We explained the damage probability in the methodology section as advised.
    - We rectified the drafting mistake.

**Discussion and conclusions**

- **Line 326: what "hydrodynamics of the period" is referred to? Is hydrodynamics changing with time? Rephrase or explain better.**

In addition to rephrasing the sentence, we are added a new sentence to provide better explanation for comprehension. We moved the sentence to the methodology section "3.1 Tsunami Wave Potential Assessment".

- **Line 362: for surface roughness, the authors used the present conditions: are they similar to 1945 ones? Using the present conditions to reconstruct a past event can produce excessive waves dampening, affecting the simulations and the source selection. In 1945 probably the "roughness" was smaller, and maybe a smaller tsunami in sufficient to produce the observed effects. Please discuss this.**

We removed the 1945 Tsunami case form this candidate publication. Therefore, this research is based on the present morphology beneficial more for today's community.

- **Lines 380-381: apart from vertical structures and sirens, also panels for evacuation routes and, most of all, education of the population is basic.**

We revised the discussion and conclusion section and emphasized the importance and awareness of education of community regarding the tsunami hazard.

**Figures and Tables**

- **Figure 1: panel A) The symbols used for landslide location are incomprehensible and unrecognisable. Landslide sources are briefly discussed but not used in the simulations, please consider also discarding them. The label Arabian Sea should be moved here from panel B. Panel B) The black line, marking the inundation limit, is not uniform but changes thickness: why?**

Panel A) Landslide location and related section are removed. The label "Arabian Sea" is moved to Panel A.

The Penel-B is removed along with exclusion of 1945 Tsunami case.

- **Table 2: why Table 2? Where is Table 1?**

The Tables sequence is revised as.

- Table 1 (in main manuscript body)
- Table 2 (in main manuscript body)
- Table 3 (in Supplement)
- Table 4 (in Supplement)

- **Figures 2, 3, 4, 5, 6, 7:**

**the toponyms are poorly readable on the map. Maybe the authors should consider putting them into a separate box with the corresponding numbers, leaving only these on the map.**

**For a better comparison between the two models, the maps should cover the same domain: as it is now, it isn't easy to assess the differences between the two approaches.**

**The graphs would benefit from adding some ticks on the horizontal axis, to see also fractions of hours.?**

We removed the static model and all its related content.

The figures are updated with respect to toponyms. We added horizontal and vertical ticks for fractional analysis along both axis of the graph.

**MINOR ISSUES AND TYPOS**

**Minor issues and typos**

All addressed as advised

---

## Editor Decision (ED1)

**Comments to the revised version – Reviewer#2**

**General comments**

The paper by Haider and coauthors has significantly improved after the revision and is now more clear and complete: I appreciated their great effort.

However, in my opinion some minor issues still need attention before publication: they are reported in the "Specific comments" section below, together with some typos in the following section.

**Specific comments**

*3.    Methods*

Line 67: "*In addition to the recorded tsunami wave height of 12-15 m in 1945 at Pasni…*" This information has not been provided before, but here is commented as already known. Maybe it should be included in the Introduction where the 1945 tsunami effects in Pasni are described, and then recalled here.

Lines 78-79: the utility of the "Fins scenario" is not totally clear: if it is used to "validate" the tsunamites, why isn't the respective flooding shown and discussed as the other scenarios? Anyway, in Table 1 it is denoted as scenario D, so for coherence it should be cited like that in the text.

Line 80: With "*The other tsunami sources*" are you referring to other kind of tsunami sources (landslide, etc), or to other potential seismic sources? This is a bit misleading.

Line 81 (see also comment of Line 67): the information about the 1945 Pasni tsunami could be moved to Introduction or can be left here but the sentence at Line 67 has to be removed.

Line 92 and Table 1: do A, B, C refer to 7m, 10m and 15m scenarios respectively? If yes, write it explicitly, or refer to these scenarios in the text accordingly.

Lines 100-101: this sentence causes a bit of confusion. Since the authors have already cited three scenarios, are these two additional?

Line 133: the assumption of 0.99 for the parameter $C_D$ should be motivated, at least with a citation.

*4. Results and interpretations*

Line 199: the authors should explain (in the previous "Methods" section) how this probability is computed and used. Is this the probability for each building to be destroyed? Or is it the percentage of destroyed buildings over the total? Are partial damages accounted for? Add a brief description of the application of the vulnerability method.

*5. Discussion and conclusions*

Lines 238-240: indeed, also wave dispersion can play a role, if the wavelength is reduced considerably and the shallow-water approximation isn't valid anymore.

Line 290: tsunamis do not manifest on the coast only by sea retreat. The polarity of the first arrival depends on many features: the source characteristics, non-linear effects in shallow water, the position of the coast with respect to the source.

**Minor issues and typos**

Line 79: "*The tsunamis ARE generated*".

Line 80: "*Table 1*" instead of "*table 1*".

Line 97: "*multiproxy proxy*" does not sound correct; remove "*is*" after "*return period*".

Line 140: "*experiences*" instead of "*experience*".

Line 143: "*In the 10 m wave scenario, along with first wave a secondary wave with a wave height of 8 m is generated*". The word "*wave*" is repeated too many times, rephrase this sentence or change wording.

Line 193: At the end of the sentence, remove ".

---

## Author Response (AR2)

**Dear Editor, Dear Anonymous Referee #2,**

We thank you for your appreciation and valuable time, effort, and insightful comments on our manuscript. Each comment and suggestion have been addressed in detail below.

Referee #2 Comments in **Orange colour**

Authors' replies in **Black**

========================================================================

**COMMENTS and REPLIES (Reviewer#2)**

**Specific comments**

*3. Methods*

Line 67: "*In addition to the recorded tsunami wave height of 12-15 m in 1945 at Pasni…*" This information has not been provided before, but here is commented as already known. Maybe it should be included in the Introduction where the 1945 tsunami effects in Pasni are described, and then recalled here.

REPLY: We included it in the Introduction part as advised (Line: 38).
* * *
Lines 78-79: the utility of the "Fins scenario" is not totally clear: if it is used to "validate" the tsunamites, why isn't the respective flooding shown and discussed as the other scenarios? Anyway, in Table 1 it is denoted as scenario D, so for coherence it should be cited like that in the text.

REPLY: Conceptually, the Fins Scenario is designed to estimate the tsunami potential in the Arabian Sea and validating it through the tsunamites morphodynamics at the Fins locality. The propagation and coastal interaction are given as the simulation file (Supplements Simulations-1 Fins-Sur, Oman) and discussed in the Methodology section. In the updated manuscript, we ensured that the Fins scenario is consistently referred to as "*Fins Scenario*" in the text for coherence as advised (Line: 78, 92).
* * *
Line 80: With "*The other tsunami sources*" are you referring to other kind of tsunami sources (landslide, etc), or to other potential seismic sources? This is a bit misleading.

REPLY: We acknowledge the textual confusion. The sources are clarified in the updated manuscript (Line: 79-80).

Now the sentence appears as "*The other tsunami sources such as landslides, volcanoes, and meteoric impacts in the ocean water…….*"
* * *
Line 81 (see also comment of Line 67): the information about the 1945 Pasni tsunami could be moved to Introduction or can be left here but the sentence at Line 67 has to be removed.

REPLY: We improved the manuscript by choosing the later choice and deleted the Line 67 as advised.
* * *
Line 92 and Table 1: do A, B, C refer to 7m, 10m and 15m scenarios respectively? If yes, write it explicitly, or refer to these scenarios in the text accordingly.

REPLY: We explicitly referred the scenarios in the Table 1 and in the text both as advised.
* * *
Lines 100-101: this sentence causes a bit of confusion. Since the authors have already cited three scenarios, are these two additional?

REPLY: We acknowledge the confusion and rectified as advised. We have rephrased the sentence to remove the confusion. Now, it appears as "In total three different scenarios have been modelled for enhanced vulnerability assessment of the study area" (Line: 100-101).
* * *
Line 133: the assumption of 0.99 for the parameter CD should be motivated, at least with a citation.

REPLY: We addressed the choice of the parameters with citation as advised (Line: 33)
* * *
*4. Results and interpretations*

Line 199: the authors should explain (in the previous "Methods" section) how this probability is computed and used. Is this the probability for each building to be destroyed? Or is it the percentage of destroyed buildings over the total? Are partial damages accounted for? Add a brief description of the application of the vulnerability method.

REPLY: We have given the relative explanation to all the parts of the question. The partial damages are not taken into the account. We have also added an example showing calculation of the probability through the hydrodynamic force (Line: 125-145).

The calculation of partial damages is not calculated as it needs further data, research and time.
* * *
*5. Discussion and conclusions*

Lines 238-240: indeed, also wave dispersion can play a role, if the wavelength is reduced considerably and the shallow-water approximation isn't valid anymore.

REPLY: Thanks for explaining the phenomenon and concept. We added the "wave dispersion phenomenon" in our sentence too (Line: 248).
* * *
Line 290: tsunamis do not manifest on the coast only by sea retreat. The polarity of the first arrival depends on many features: the source characteristics, non-linear effects in shallow water, the position of the coast with respect to the source.

REPLY: We added a sentence, technically explaining the coastal retreat or drawdown phenomenon and linked its dependency with respect to watch and alertness warning (Line 300-302).
* * *
**Minor issues and typos**

Line 79: "*The tsunamis ARE generated*".

Line 80: "*Table 1*" instead of "*table 1*". Line 97: "*multiproxy proxy*" does not sound correct; remove "*is*" after "*return period*". Line 140: "*experiences*" instead of "*experience*". Line 143: "*In the 10 m wave scenario, along with first wave a secondary wave with a wave height of 8 m is generated*". The word "*wave*" is repeated too many times, rephrase this sentence or

change wording. Line 193: At the end of the sentence, remove ".

REPLY: We rectified all these errors as advised.
* * *
Title Adjustment:

We removed the "Integrated" word from the title. While undergoing major revision, we removed the static analysis component from the manuscript. Therefore, we realized the term "integration" remains no more relevant.

Now the title appears as "Tsunami inundation and vulnerability analysis at the Makran Coast, Pakistan"